# CSF Aβ42 and Aβ42/Aβ40 Ratio in Alzheimer’s Disease and Frontotemporal Dementias

**DOI:** 10.3390/diagnostics13040783

**Published:** 2023-02-19

**Authors:** Vasilios C. Constantinides, George P. Paraskevas, Fotini Boufidou, Mara Bourbouli, Efstratios-Stylianos Pyrgelis, Leonidas Stefanis, Elisabeth Kapaki

**Affiliations:** 1First Department of Neurology, School of Medicine, National and Kapodistrian University of Athens, Eginition Hospital, Vass. Sophias Ave. 74, 11528 Athens, Greece; 2Neurochemistry and Biological Markers Unit, First Department of Neurology, School of Medicine, National and Kapodistrian University of Athens, Eginition Hospital, Vass. Sophias Ave. 74, 11528 Athens, Greece; 3Second Department of Neurology, School of Medicine, National and Kapodistrian University of Athens, “Attikon” University General Hospital, Rimini 1, 12462 Athens, Greece

**Keywords:** cerebrospinal fluid, amyloid beta with 42 amino acids, tau proteins, Alzheimer’s disease, frontotemporal dementia

## Abstract

Background: Alzheimer’s disease dementia (ADD) may manifest with atypical phenotypes, resembling behavioral variant frontotemporal dementia (bvFTD) and corticobasal syndrome (CBS), phenotypes which typically have an underlying frontotemporal lobar degeneration with tau proteinopathy (FTLD-tau), such as Pick’s disease, corticobasal degeneration (CBD), progressive supranuclear palsy (PSP), or FTLD with TDP-43 proteinopathy (FTLD-TDP). CSF biomarkers total and phosphorylated tau (τ_T_ and τ_P-181_), and amyloid beta with 42 and 40 amino acids (Aβ_42_ and Aβ_40_) are biomarkers of AD pathology. The primary aim of this study was to compare the diagnostic accuracy of Aβ_42_ to Aβ_42_/Aβ_40_ ratio in: (a) differentiating ADD vs. frontotemporal dementias; (b) patients with AD pathology vs. non-AD pathologies; (c) compare biomarker ratios and composite markers to single CSF biomarkers in the differentiation of AD from FTD; Methods: In total, 263 subjects were included (ADD: *n* = 98; bvFTD: *n* = 49; PSP: *n* = 50; CBD: *n* = 45; controls: *n* = 21). CSF biomarkers were measured by commercially available ELISAs (EUROIMMUN). Multiple biomarker ratios (Aβ_42_/Aβ_40_; τ_T_/τ_P-181_; τ_T_/Aβ_42_; τ_P-181_/Aβ_42_) and composite markers (t-tau: τ_T_/(Aβ_42_/Aβ40); p-tau: τ_P-181_/(Aβ_42_/Aβ_40_) were calculated. ROC curve analysis was performed to compare AUCs of Aβ_42_ and Aβ_42_/Aβ_40_ ratio and relevant composite markers between ADD and FTD, as defined clinically. BIOMARKAPD/ABSI criteria (abnormal τ_T_, τ_P-181_ Aβ_42_, *and* Aβ_42_/Aβ_40_ ratio) were used to re-classify all patients into AD pathology vs. non-AD pathologies, and ROC curve analysis was repeated to compare Aβ_42_ and Aβ_42_/Aβ_40_; Results: Aβ_42_ did not differ from Aβ_42_/Aβ_40_ ratio in the differentiation of ADD from FTD (AUCs 0.752 and 0.788 respectively; *p* = 0.212). The τ_T_/Aβ_42_ ratio provided maximal discrimination between ADD and FTD (AUC:0.893; sensitivity 88.8%, specificity 80%). BIOMARKAPD/ABSI criteria classified 60 patients as having AD pathology and 211 as non-AD. A total of 22 had discrepant results and were excluded. Aβ_42_/Aβ_40_ ratio was superior to Aβ_42_ in the differentiation of AD pathology from non-AD pathology (AUCs: 0.939 and 0.831, respectively; *p* < 0.001). In general, biomarker ratios and composite markers were superior to single CSF biomarkers in both analyses. Conclusions: Aβ_42_/Aβ_40_ ratio is superior to Aβ_42_ in identifying AD pathology, irrespective of the clinical phenotype. CSF biomarker ratios and composite markers provide higher diagnostic accuracy compared to single CSF biomarkers.

## 1. Introduction

Over the past three decades, advances in cerebrospinal fluid (CSF) and positron emission tomography (PET) biomarkers have been the defining factor in evolving the conceptual framework of Alzheimer’s disease (AD) from a simple clinical entity, characterized by amnestic-predominant dementia, to a biological-clinical continuum, with diverse clinical manifestations [1]. The defining neuropathological lesions of AD are amyloid plaques (extracellular accumulation of pathologically misfolded β amyloid) and neurofibrillary tangles (consisting of hyper-phorphorylated tau protein), which result in neurodegeneration [2,3].

The AT(N) system has supported the classification of biomarkers of diverse modalities (CSF, PET, or MRI) into three groups: (A) for amyloidosis, (T) for tau-pathology, and (N) for neurodegeneration [4]. Within this framework, total tau protein (τ_Τ_), phosphorylated tau protein at threonine 181 (τ_P-181_), and amyloid beta with 42 amino acids (Aβ_42_) have been classified as markers of neurodegeneration, tau-pathology, and amyloidosis, respectively.

A decrease in CSF Aβ_42_ is characteristic of AD. However, due to significant inter-subject variability in Aβ_42_ levels, defining an Aβ_42_ cut-off with high diagnostic accuracy for discrimination between AD and non-AD pathologies has been problematic [5]. Moreover, Aβ_42_ measurement is particularly sensitive to alterations in pre-analytical factors [6,7]. Several studies have supported that the incorporation of CSF amyloid beta with 40 amino acids (Aβ_40_), which is a reflection of total CSF amyloid levels, by use of the Aβ_42_/Aβ_40_ ratio, is a better marker of the relatively selective decrease in Aβ_42_ in AD [8,9].

Most of the studies comparing the diagnostic accuracy of Ab_42_ to Aβ_42_/Aβ_40_ ratio have defined AD or other dementias by use of clinical criteria [10,11,12,13,14,15,16,17,18,19,20]. However, this approach is problematic, since AD may manifest with atypical non-amnestic presentations, including a language presentation (i.e., logopenic variant primary progressive aphasia), a visuospatial presentation (i.e., posterior cortical atrophy), a dysexecutive presentation (mimicking behavioral variant of frontotemporal dementia) and corticobasal syndrome [21,22]. Thus, the use of clinical criteria to define AD will result in the misclassification of AD patients as non-AD in cases of atypical manifestations and vice versa.

Several studies have compared Aβ_42_ and Aβ_42_/Aβ_40_ ratio to amyloid-PET, in an attempt to investigate the optimal CSF amyloid marker [8,9,23,24,25]. Most of these studies conclude that the Aβ_42_/Aβ_40_ ratio results in higher concordance with amyloid-PET compared to Aβ_42_ [8,9,24,25], although a single study did not report a difference between the two markers [23]. However, most of these studies have only included healthy subjects or patients with mild cognitive impairment or dementia due to AD, without the inclusion of other dementias. To date, a study comparing CSF Aβ_42_ to Aβ_42_/Aβ_40_ in a cohort with neuropathological confirmation of clinical diagnoses is lacking.

The present study aimed to compare the predictive values of Aβ_42_ and Aβ_42_/Aβ_40_ ratio for an underlying AD pathology in a cohort of patients with diverse dementing disorders. For the purposes of this study, we selected to include patients with a clinical diagnosis of AD dementia (ADD) and various frontotemporal dementias (FTD), including behavioral variant FTD (bvFTD), progressive supranuclear palsy (PSP) and corticobasal degeneration (CBD), as well as healthy subjects. We opted not to include patients with Lewy body dementia, due to the high incidence of co-occurrence of AD in these patients, rendering interpretation of CSF biomarkers problematic in the absence of biomarkers in other modalities (i.e., PET-CT).

Initially, the diagnostic accuracies of Aβ_42_ and Aβ_42_/Aβ_40_ as predictors of *AD dementia* were compared among study groups based on clinical diagnoses, in accordance with the methodology applied in most relevant studies. We then applied the BIOMARKAPD/ABSI criteria in all patients, irrespective of their clinical phenotype, and re-classified them as having an AD or a non-AD underlying pathology [26,27]. CSF Aβ_42_ and Aβ_42_/Aβ_40_ were subsequently re-applied in this setting, in order to compare their diagnostic accuracy for underlying *AD pathology*.

To further compare Aβ_42_ and Aβ_42_/Aβ_40_, we included several composite markers in our analyses, by replacement of Aβ_42_ with Aβ_42_/Aβ_40_ in the τ_T_/Aβ_42_ and τ_P-181_/Aβ_42_ ratios.

## 2. Materials and Methods

### 2.1. Patients

The medical files of all patients with available data on CSF biomarkers Aβ_42_, Aβ_40_, τ_T_, and τ_P-181_, who were admitted from 2011 to 2021 to the “Neurodegenerative Disorders and Epilepsy” Ward of our hospital, were retrospectively reviewed. For the purposes of this study, subjects were included if they fulfilled the established diagnostic criteria for the following diseases: (a) ADD [21]; (b) bvFTD [28]; (c) PSP [29], and (d) CBD [30]. For comparison reasons, a control group was included. This consisted of otherwise healthy subjects, with no comorbidities, undergoing knee or hip joint surgery or hernia repair under spinal anesthesia. These subjects had a negative history of cognitive or behavioral/psychiatric disorders and no clinical evidence of any major disease. All subjects had normal scores on neuropsychological testing (Mini Mental State Examination and Frontal Assessment Battery) [31,32].

### 2.2. CSF Sampling and Biomarker Measurements

All patients underwent lumbar puncture at 10–11 a.m., after overnight fasting, based on standard operating procedures in accordance to recommendations to standardize pre-analytical confounding factors in AD CSF biomarkers [33].

CSF biomarkers Aβ_42_, Aβ_40_, τ_T_, and τ_P-181_ were measured in duplicate with ELISA by commercially available kits (EUROIMMUN Beta-Amyloid (1–42) ELISA; EUROIMMUN Beta-Amyloid (1–40) ELISA; EUROIMMUN Total-Tau ELISA; EUROIMMUN pTau (181) ELISA respectively), according to manufacturer instructions.

Our laboratory implements both internal and external quality control measures to ensure the accuracy of measurements longitudinally. Specifically, for internal control a pooled CSF sample is used in every test run, resulting in an over >90% between-run precision. As for external control, we participate in “The Alzheimer’s Association’s QC program”, which provides additional external pooled CSF samples for validating results reliability regardless of the kit’s lot number.

Additionally, the following CSF biomarker ratios, which incorporate two CSF AD biomarkers were calculated: Aβ_42_ to Aβ_40_ (Aβ_42_/Aβ_40_), τ_T_ to τ_P-181_ (τ_T_/τ_P-181_), τ_T_ το Aβ_42_ (τ_T_/Aβ_42_) and τ_P-181_ to Aβ_42_ (τ_P-181_/Aβ_42_). All of these ratios have been previously applied in studies in an effort to increase the diagnostic accuracy of CSF biomarkers.

Lastly, in an effort to incorporate two CSF AD biomarkers in a single marker, the following composite markers were calculated: (a)Composite t-tau marker: τ_T_/(Aβ_42_/Aβ_40_).

The τ_T_/Aβ_42_ ratio has been previously applied as an AD neurochemical marker, based on the observed increase in τ_T_ and decrease in Aβ_42_ in patients with an underlying AD pathology. Several studies support that the Aβ_42_/Aβ_40_ ratio may provide improved diagnostic accuracy for amyloid pathology compared to Aβ_42_. To look into this hypothesis, we introduced this composite marker.

(b)Composite p-tau marker: τ_P-181_/(Aβ_42_/Aβ_40_).

The τ_P-181_/Aβ_42_ ratio has been previously applied as an AD neurochemical marker, based on the observed increase in τ_P-181_ and decrease in Aβ_42_ in patients with an underlying AD pathology. As mentioned previously, substituting Aβ_42_ with the Aβ_42_/Aβ_40_ ratio may provide improved diagnostic accuracy for amyloid pathology. Composite p-tau marker:

### 2.3. Ethical Considerations

All patients or their next of kin (in cases of compromised mental capacity) provided written informed consent for participation in this study. The study was approved by the Scientific and Ethics Committee of Eginition Hospital and was performed in accordance with the guidelines of the 1964 Declaration of Helsinki.

### 2.4. Statistical Analysis

The normality of distribution and homogeneity of variances were checked by Shapiro–Wilk’s and Levene’s tests, respectively. Comparison of clinical, neuropsychological, and CSF biomarker characteristics between study groups was performed by ANOVA (with Bonferroni correction for multiple comparisons) or Kruskal–Wallis test as appropriate.

We performed two sets of analyses. The initial analysis was based on the clinical diagnoses of the study subjects. Thus, Receiver Operating Characteristic (ROC) Curve analysis was performed to compare the diagnostic accuracy of all CSF biomarkers, biomarker ratios, and composite biomarkers in differentiating between patients with *AD dementia* vs. all other clinical groups. Area under the curve (AUC), 95% confidence interval of the AUC, cut-off point with optimal diagnostic accuracy (defined as maximal sensitivity and specificity), as well as specificity, sensitivity, and Youden Index (YI) of optimal cut-off points, were calculated.

In order to look into possible differences between AUCs of ROC curves of various biomarkers in the identification of AD dementia, the De Long method was applied. In an effort to compare the diagnostic accuracy of Aβ_42_ vs. Aβ_42_/Aβ_40_ ratio, the following comparisons of ROC curves were performed: (a) Aβ_42_ vs. Aβ_42_/Aβ_40_ ratio; (b) τ_T_/Aβ_42_ vs. composite t-tau: (c) τ_P-181_/Aβ_42_ vs. composite p-tau.

The second analysis aimed to investigate the diagnostic accuracy of single CSF biomarkers, biomarker ratios, and composite biomarkers in identifying *AD pathology* irrespective of the clinical phenotype. To this end, a two-step process was applied, as described elsewhere.

Initially, CSF biomarkers were transformed into binary variables (i.e., normal or abnormal), based on cut-off values of the Unit of Neurochemistry and Biomarkers (Aβ_42_ < 480 pg/mL; τ_T_ > 400 pg/mL; τ_P-181_ > 60 pg/mL; Aβ_42_/Aβ_40_ < 0.094), as described previously [34]. Subjects with abnormal Aβ_42_, τ_T_, τ_P-181_, and Aβ_42_/Aβ_40_ ratio (A+T+N+) were considered to harbor AD pathology, based on the BIOMARKAPD/ABSI criteria [26,27]. Thus, all patients, irrespective of their clinical phenotype were classified into two categories: AD vs. nonAD.

We elected to apply the most stringent classification criterion (i.e., *all three CSF biomarkers abnormal*) for AD pathology identification in order to increase specificity, at the expense of sensitivity. For amyloid pathology (A) in particular, there were two available established CSF markers: Aβ_42_ and the Aβ_42_/Aβ_40_ ratio. In an effort to increase specificity, subjects with a decrease in both markers (Aβ_42_ and Aβ_42_/Aβ_40_) were considered A(+), and only subjects with normal values in both biomarkers were considered A(−). Of the 263 subjects, 22 (8.4%; AD dementia: 16 patients; CBD: 3 patients; PSP: 1 patients; controls: 2 subjects) had discrepant results in amyloid pathology identification based on Aβ_42_ and Aβ_42_/Aβ_40_ ratio and were excluded from this analysis.

Thus, this second analysis included 82 patients with AD pathology and 140 patients with non-AD pathology: 49 bvFTD patients, 42 CBD patients, and 49 PSP patients. The majority of the non-AD pathology patients are considered to have an underlying frontotemporal lobar degeneration (FTLD), with most PSP and CBD patients harboring an FTLD with tau proteinopathy (FTLD-tau) and most bvFTD patients harboring either an FTLD-tau or FTLD-TDP43 proteinopathy. Thus, in essence, the second analysis referred to a comparison of Aβ_42_ to Aβ_42_/Aβ_40_ ratio in the differentiation of AD pathology from FTLD. However, due to the lack of neuropathological confirmation, we elected to use the term non-AD pathology instead of FTLD, since the presumed underlying pathology is based on CSF biomarker profiles.

Following this classification of all subjects irrespective of their phenotype into the AD and nonAD groups, ROC curve analysis was applied to determine the discriminative power of CSF biomarkers, biomarker ratios, and composite markers for this differentiation. Cut-off points were determined based on the maximal combined sensitivity and specificity criterion. Area under the curve (AUC), 95% confidence interval of the AUC, YI, sensitivity, and specificity were also calculated.

In order to look into possible differences between AUCs of ROC curves of various biomarkers in AD pathology identification, the De Long method was applied. In an effort to compare the diagnostic accuracy of Aβ_42_ vs. Aβ_42_/Aβ_40_ ratio, the following comparisons of ROC curves were performed: (a) Aβ_42_ vs. Aβ_42_/Aβ_40_ ratio; (b) τ_T_/Aβ_42_ vs. composite t-tau: (c) τ_P-181_/Aβ_42_ vs. composite p-tau.

All analyses were performed by IBM SPSS Statistics^®^ version 23.0.0.0 (SPSS Inc., Chicago, IL, USA, 2013). All graphs were designed using GraphPad Prism^®^, version 5.03 (GraphPad Software Inc., La Jolla, CA, USA, 2009).

## 3. Results

### 3.1. Clinical and Demographic Data

In total, 263 subjects were included (ADD: 98 patients; bvFTD: 49 patients; PSP: 50 patients; CBD: 45 patients; controls: 21 subjects). Study groups differed in age (*p* = 0.003), with the control group exhibiting the greatest age compared to other groups and sex distribution (*p* = 0.005). Control subjects performed significantly better in the MMSE and FAB tests compared to patient groups, as expected (*p* < 0.001) (Table 1).

### 3.2. Comparison of CSF Biomarkers, Biomarker Ratios, and Composite Markers with ANOVA

Study groups differed significantly in all CSF biomarkers, biomarker ratios, and composite markers, with ADD patients exhibiting higher values in τ_T_, τ_P-181_, τ_P-181_/τ_T_ ratio, τ_T_/Aβ_42_ ratio, τ_P-181_/Aβ_42_ ratio, and in all composite markers. Additionally, ADD patients exhibited the lowest values of Aβ_42_ and Aβ_42_/Aβ_40_ ratio as expected *p* < 0.001 for all comparisons) (Table 1, Figure 1).

### 3.3. ROC Curve Analysis of CSF Biomarkers for the Differentiation between Alzheimer’s Disease Dementia vs. All Other Clinical Groups

The composite t-tau and p-tau markers provided comparable high diagnostic accuracies (AUCs: 0.868 and 0.869, respectively). Regarding CSF biomarker ratios, the τ_T_/Aβ_42_ and τ_P-181_/Aβ_42_ ratios resulted in higher AUCs (0.893 and 0.878, respectively) compared to Aβ_42_/Aβ_40_ and τ_P-181_/τ_T_ (0.788 and 0.755, respectively). Among single CSF biomarkers, τ_P-181_ provided maximal AUC, followed by τ_T_ and Aβ_42_ (0.886, 0.840, and 0.752, respectively) (Table 2) (Figure 2a).

### 3.4. Comparison of ROC Curves Related to Aβ_42_ vs. Aβ_42_/Aβ_40_ Ratio for ADD vs. Other Clinical Groups Differentiation

There were no statistically significant differences between Aβ_42_ vs. Aβ_42_/Aβ_40_ (AUC 0.752 vs. 0.788, respectively), τ_Τ_/Aβ_42_ vs. τ_T_/(Aβ_42_/Aβ_40_) (AUC 0.893 vs. 0.868), respectively) or τ_P-181_/Aβ_42_ vs. τ_P-181_/(Aβ_42_/Aβ_40_) (AUC 0.878 vs. 0.869, respectively) in differentiating ADD from other clinical syndromes (Table 3).

### 3.5. ROC Curve Analysis of CSF Biomarkers for the Differentiation between AD vs. Non-AD Pathology, Irrespective of Clinical Phenotype

The composite t-tau marker provided a maximal AUC of 0.985 for the differentiation of AD pathology from non-AD pathologies, with a cut-off of >5.82 resulting in a 100% sensitivity and 91.2 specificity. The p-tau markers provided comparable AUCs of 0.965. Among CSF biomarker ratios, the τ_T_/Aβ_42_ resulted in maximal AUC (0.975) followed by τ_P-181_/Ab_42_, Aβ_42_/Aβ_40_, and τ_P-181_/τ_T_ ratios (AUCs: 0.952, 0.939, and 0.780, respectively). Among single CSF biomarkers, τ_P-181_ resulted in optimal AUC (0.960) followed by τ_T_ and Aβ_42_ (AUC 0.948 and 0.831 respectively) (Table 4).

### 3.6. Comparison of ROC Curves Related to Aβ_42_ vs. Aβ_42_/Aβ_40_ Ratio for AD vs. Non-AD Pathology Differentiation

There were no statistically significant differences between τ_Τ_/Aβ_42_ vs. τ_T_/(Aβ_42_/Aβ_40_) (AUC 0.975 vs. 0.985), respectively) or τ_P-181_/Aβ_42_ vs. τ_P-181_/(Aβ_42_/Aβ_40_) (AUC 0.952 vs. 0.965, respectively) in differentiating AD pathology from non-AD pathology. However, Aβ_42_/Aβ_40_ differed significantly from Aβ_42_ in differentiating AD from non-AD pathologies (AUC 0.831 vs. 0.939, respectively; *p* < 0.001) (Table 3).

## 4. Discussion

The primary aim of this study was to compare the predictive value of Aβ_42_/Aβ_40_ ratio to Aβ_42_ in identifying: (a) *Alzheimer’s disease dementia* (ADD) from other clinical phenotypes and (b) *Alzheimer’ disease pathology* from non-AD pathologies. Regarding the clinical distinction of ADD from other phenotypes, the Aβ_42_/Aβ_40_ ratio and Aβ_42_ provided comparable diagnostic accuracy in our cohort. In agreement with our study, most relevant studies in the literature relying on clinical criteria for AD definition support that the Aβ_42_/Aβ_40_ ratio provides greater AUC values compared to Aβ_42_ in diagnosing ADD [12,13,14,15,16,17,18,19,20]. Few studies, however, have conflicting results, supporting the superiority of Aβ_42_ over theAβ_42_/Aβ_40_ ratio or no difference between the two markers [10,11]. Importantly, contrary to our study, these studies do not include statistical comparisons of AUC values and rely on numerical differences between AUCs.

Importantly, the Aβ_42_/Aβ_40_ ratio provided significantly greater diagnostic accuracy compared to Aβ_42_ in the differentiation of AD pathology from non-AD pathologies. Likewise, Aβ_42_/Aβ_40_-derived composite markers produced greater AUCs το Aβ_42_-derived ratios, although these differences did not reach statistical significance. Due to the inclusion in the present study of phenotypes that typically have an underlying FTLD pathology, either with tau proteinopathy (FTLD-tau) or TDP-43 (FTLD-TDP), this finding signifies the importance of the Aβ_42_/Aβ_40_ ratio in differentiating between AD pathology and FTLD. However, this finding needs verification by studies with both CSF biomarker and neuropathological data available. This finding is particularly important from a clinical and research perspective because it highlights the importance of the Aβ_42_/Aβ_40_ ratio in the in vivo identification of amyloid pathology in patients with diverse clinical phenotypes. Μost studies comparing CSF biomarkers to amyloid-PET support this finding [8,9,24,25], although a study reported conflicting results, depending on the assay used [23].

A third finding of our study is the significant disparity between clinical diagnosis and underlying pathology, highlighting the significant clinical heterogeneity of AD [21,22] as well as the significant pathological heterogeneity of phenotypes such as CBS [35,36]. Multiple clinical—pathological studies have supported this clinical—pathological overlap [37,38,39,40]. However, this disparity was particularly high for ADD patients in our cohort (47% of ADD patients had a CSF AD profile, 16% had discrepant results and 37% had non-AD pathologies) and exceedingly low for bvFTD (all patients had non-AD CSF profile).

This finding can be attributed to the criteria we applied for the definition of the CSF AD biochemical profile. We elected for the purposes of this study to define the AD CSF profile according to the BIOMARKAPD/ABSI criteria, which require all three major biomarker groups (amyloid, tau, and neurodegeneration) to be abnormal. Additionally, we elected to define as A(+) patients with abnormal values in both Aβ_42_ and Aβ_42_/Aβ_40_ ratio. These stringent criteria were applied to enhance the specificity of AD classification at the expense of sensitivity, in the absence of other biomarkers such as amyloid PET, and with no neuropathological data available. Thus, the classification of patients in AD or non-AD pathologies do not reflect the routine clinical practice, wherein a single amyloid CSF biomarker is sufficient to characterize the (A) status in the ATN system, and patients with A(+)T(+)N(−) are classified as AD. To support this hypothesis, when applying the ATN criteria based on Aβ_42_/Aβ_40_ ratio, 72% of patients were classified as AD pathology (data not shown).

In CSF biomarker studies looking into inherent differences among neurodegenerative disorders, the significant clinical—pathological overlap in neurodegenerative disorders can be overcome if a two-step algorithm is applied, as proposed previously [41,42]. As a first step, all patients, irrespective of their clinical phenotype, should initially be classified as AD or non-AD based on a biomarker taxonomic system (e.g., ATN or BIOMARKAPD/ABSI). This will assist in identifying atypical presentations of AD (e.g., PPA, behavioral-frontal dementia, CBS, etc.) and avoid misclassification of these subjects. The second step would involve the direct comparison of biomarker levels among different disorders.

An interesting finding was the improved diagnostic accuracy of CSF biomarker ratios and composite markers compared to single CSF biomarkers. The τ_T_/Aβ_42_ ratio and the composite markers (t-tau and p-tau composite marker) provided excellent discriminative power for AD pathology identification. Although these ratios and composite markers lack inherent biological meaning, the incorporation of two biomarkers (in the case of CSF biomarker ratios) or three biomarkers (in the case of composite markers) in a single continuous variable for each patient increases discriminative power compared to single CSF biomarkers. To this extent, CSF biomarker ratios have been previously implemented and have yielded excellent concordance with neuropathological data in FTD [43,44]. Additionally, CSF ratios were superior to single CSF biomarkers in the differential diagnosis of AD from bvFTD [43,45,46,47].

An additional theoretical advantage of using CSF biomarker ratios and composite markers instead of single CSF biomarkers is that they may assist in decreasing the inter-subject variability in pre-analytical variables, such as type of sampling and storage tubes, incubation time, number of freeze/thaw cycles, type of pipettes used, etc. This has previously been proven for the Aβ_42_/Aβ_40_ ratio [6,7].

The present study has certain limitations. Firstly, our cohort, as is the case with all relevant studies in the literature, lacks neuropathological confirmation, which is the golden standard for diagnosis in neurodegenerative disorders. For this reason, we only included well-characterized patients based on the most recently established diagnostic criteria with adequate follow-up. A second limitation is the absence of a biomarker of a different modality (e.g., amyloid PET), to use as a reference. However, PET-CT can only provide information on a single axis of the ATN triad. For this reason, we selected to apply the BIOMARKAPD/ABSI criteria, to classify patients into AD or non-AD pathologies. Additionally, in order to minimize the effect of cyclical error in comparing Aβ_42_ το Aβ_42_/Aβ_40_, we used all four major AD CSF biomarkers for AD classification (Aβ_42_, Aβ_42_/Aβ_40_ ratio, τ_T_, and τ_P-181_).

Our study, in accordance with the literature, supports the improved diagnostic accuracy of the Aβ_42_/Aβ_40_ ratio compared to Aβ_42_ in identifying AD pathology. Moreover, we highlighted the exceptionally high diagnostic accuracy of composite markers. Further studies, with neuropathological confirmation, are needed, to establish the correlation between CSF biomarkers and neuropathological characteristics.

## Figures and Tables

**Figure 1 diagnostics-13-00783-f001:**
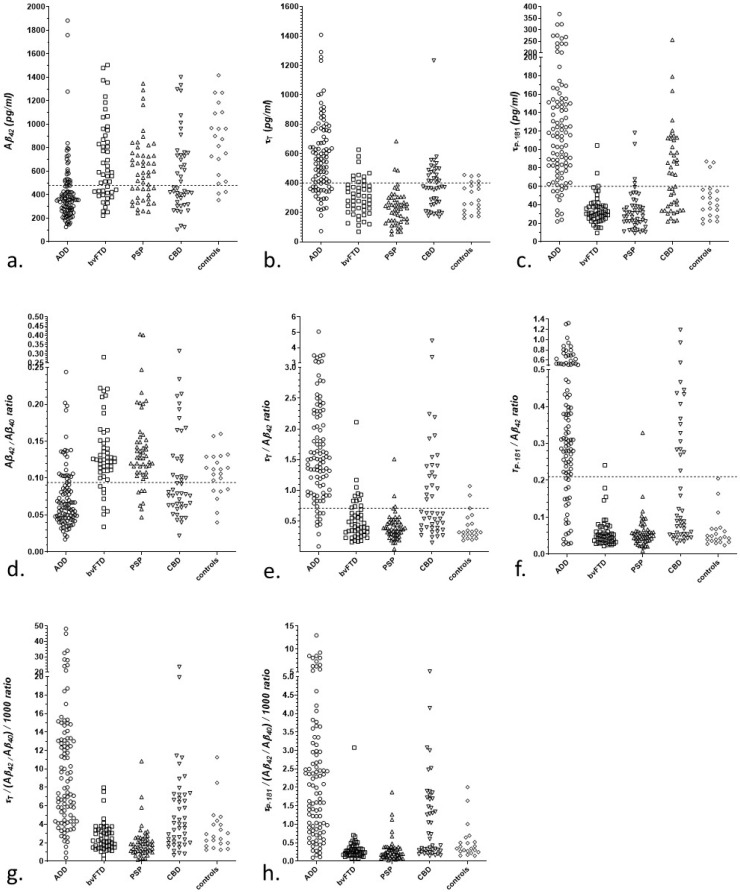
Scatterplots of CSF biomarkers (**a**–**c**), biomarker ratios (**d**–**f**) and composite markers (**g**,**h**) in study groups. Dotted lines represent normal values of our Laboratory. Data are presented for Aβ_42_ (**a**); τ_T_ (**b**); τ_P-181_ (**c**); Aβ_42_/Aβ_40_ ratio (**d**); τ_T_/Aβ_42_ ratio (**e**); _τP-181_/Aβ_42_ ratio (**f**); composite t-tau marker (τ_T_/(Aβ_42_/Aβ_40_) (**g**); composite ph-tau marker (τ_P-181_/(Aβ_42_/Aβ_40_) (**h**).

**Figure 2 diagnostics-13-00783-f002:**
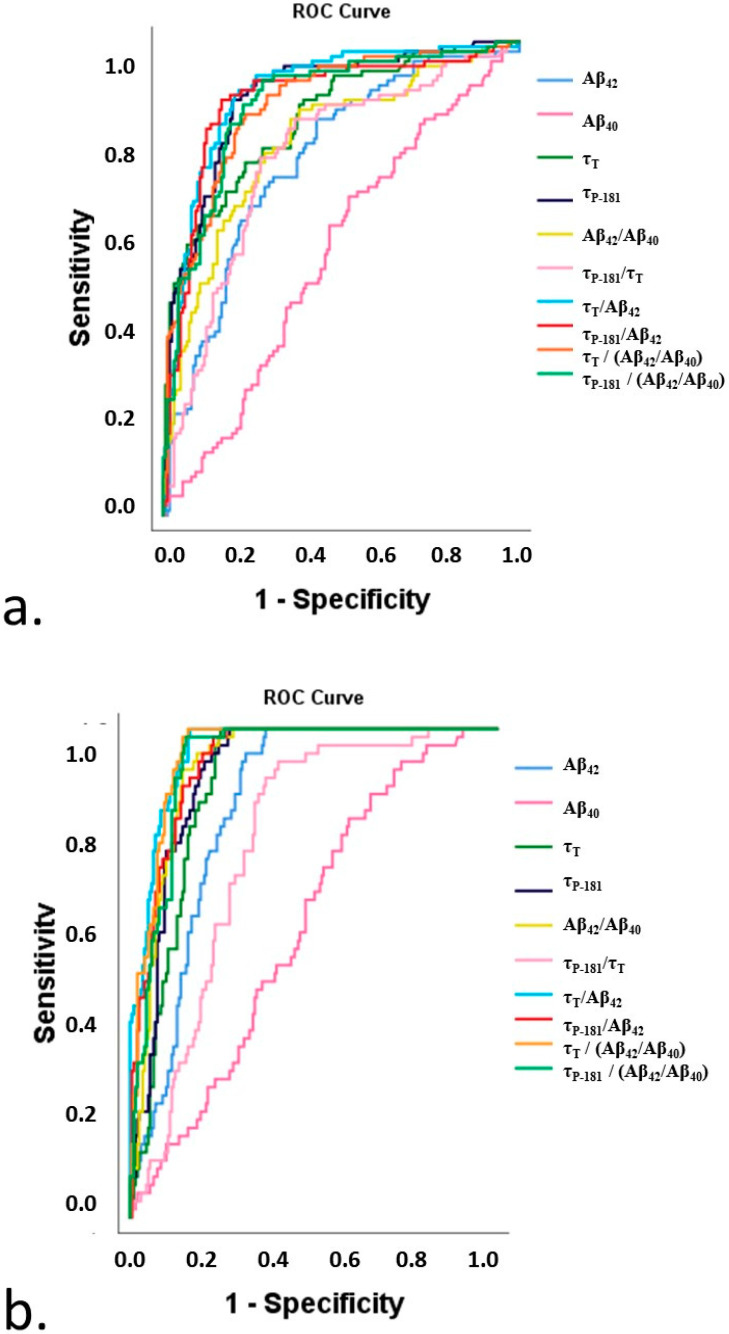
ROC curves of CSF biomarkers, biomarker ratios, and composite markers in the differentiation of: (**a**) Alzheimer’s disease dementia vs. all other clinical phenotypes and (**b**) Alzheimer’s disease pathology from non-Alzheimer’s disease pathologies, irrespective of the clinical phenotype.

**Table 1 diagnostics-13-00783-t001:** Demographic, clinical, and CSF biomarker data. ADD: Alzheimer’s disease dementia; bvFTD: behavioral variant frontotemporal dementia; PSP: progressive supranuclear palsy; CBD: corticobasal degeneration; MMSE: Mini Mental State Examination; FAB: Frontal Assessment Battery. All data are presented as mean (SD) or median (25th–75th quartile); *: x2 test; †: Kruskal-Wallis test; ‡: ANOVA.

	ADD *n* = 98	bvFTD *n* = 49	PSP *n* = 50	CBD *n* = 45	Controls *n* = 21	*p* Value
Demographic/clinical data						
Gender (m/f)	38/60	28/21	28/28	21/24	3/18	0.005 *
Age (y)	65 (57–72)	63 (58–70)	65.5 (60–69)	65 (60–74)	74 (69–76)	0.003 †
Age at onset (y)	58.5 (53–70)	61 (55–66)	62 (57–67)	62.5 (57.5–71)	NA	0.413 †
Disease duration (m)	36 (24–48)	30 (24–48)	30 (18–48)	24 (18–48)	NA	0.593 †
Education (y)	12 (6–16)	12 (9–15)	12 (6–15)	9 (6–15.5)	14.5 (12–17)	0.720 †
MMSE	17 (13–21)	23 (16–28)	27 (23–29)	22 (17–26)	29 (29–30)	<0.001 †
FAB	9 (5–11)	9 (6–14)	11 (8–14)	9 (6–12)	15 (14–16)	<0.001 †
BIOMARKAPD/ABSI Classification (*n*/%)						
AD	46 (47%)	0 (0%)	1 (2%)	12 (27%)	1 (5%)	
non-AD	36 (37%)	49 (100%)	48 (96%)	30 (67%)	48 (86%)	
discrepancy	16 (16%)	0 (0%)	1 (2%)	3 (6%)	2 (9%)	
CSF biomarkers						
Aβ_42_	421 (274)	697 (342)	616 (268)	577 (333)	846 (311)	<0.001 ‡
Aβ_40_	6215 (2795)	5630 (2648)	4900 (2545)	5975 (3083)	8221 (2875)	<0.001 ‡
τ_T_	577 (255)	305 (123)	233 (120)	384 (176)	304 (102)	<0.001 ‡
τ_P-181_	115 (81–151)	31 (25–39)	30 (21–39)	73 (34–103)	42 (29–55)	<0.001 †
CSF biomarker ratios						
Aβ_42_/Aβ_40_	0.075 (0.042)	0.132 (0.049)	0.142 (0.068)	0.105 (0.062)	0.106 (0.030)	<0.001 ‡
τ_P-181_/τ_T_	0.217 (0.183–0.263)	0.120 (0.084–0.164)	0.137 (0.106–0.176)	0.186 (0.124–0.252)	0.144 (0.126–0.153)	<0.001 †
τ_T_/Aβ_42_	1.50 (0.98–2.22)	0.44 (0.32–0.69)	0.38 (0.28–0.49)	0.63 (0.40–1.29)	0.32 (0.27–0.46)	<0.001 †
τ_P-181/_Aβ_42_	0.312 (0.223–0.514)	0.049 (0.035–0.072)	0.051 (0.037–0.063)	0.098 (0.054–0.305)	0.047 (0.036–0.068)	<0.001 †
CSF composite markers						
τ_T_/(Aβ_42_/Aβ_40_)/1000	8.34 (4.67–13.18)	2.25 (1.53–3.41)	1.75 (1.08–2.45)	4.01 (2.14–7.18)	2.67 (1.93–3.98)	<0.001 †
τ_P-181/_(Aβ_42_/Aβ_40_)/1000	1.92 (0.91–2.98)	0.26 (0.17–0.37)	0.22 (0.13–0.36)	0.73 (0.30–1.70)	0.34 (0.27–0.52)	<0.001 †

**Table 2 diagnostics-13-00783-t002:** ROC curve analysis of the diagnostic accuracy of CSF biomarkers, biomarker ratios, and composite markers in the differentiation of *Alzheimer’s disease dementia* from all other clinical syndromes and control subjects. AUC: area under the curve; 95% CI: 95% confidence interval; YI: Youden Index.

	AUC	95% CI	*p*-Value	YI	Cut-Off	Sens	Spec
CSF biomarkers							
Aβ_42_	0.752	0.695–0.803	<0.0001	0.411	<408.5	65.3	75.8
τ_Τ_	0.840	0.790–0.882	<0.0001	0.518	>455.4	63.3	88.5
τ_P-181_	0.886	0.841–0.922	<0.0001	0.684	>60.3	87.8	80.6
CSF biomarker ratios							
Aβ_42_/Aβ_40_	0.788	0.734–0.836	<0.0001	0.481	<0.096	76.5	71.5
τ_P-181_/τ_T_	0.755	0.698–0.806	<0.0001	0.482	>0.181	75.5	72.7
τ_T_/Aβ_42_	0.893	0.850–0.928	<0.0001	0.688	>0.75	88.8	80.0
τ_P-181_/Aβ_42_	0.878	0.832–0.915	<0.0001	0.714	>0.122	87.8	83.6
CSF composite markers							
τ_T_/(Aβ_42_/Aβ_40_)/1000	0.868	0.821–0.907	<0.0001	0.617	>3.96	84.7	77.0
τ_P-181_/(Aβ_42_/Aβ_40_)/1000	0.869	0.822–0.908	<0.0001	0.643	>0.55	86.7	77.6

**Table 3 diagnostics-13-00783-t003:** Comparison of ROC curve AUCs by the De Long method. AUC: area under the curve; 95% CI: 95% confidence interval.

	AUC Difference	95% CI	*p*-Value
AD dementia vs. other clinical syndromes			
Aβ_42_ vs. Aβ_42_/Aβ_40_	0.036	−0.0208–0.0934	0.212
τ_Τ_/Aβ_42_ vs. τ_T_/(Aβ_42_/Aβ_40_)	0.025	−0.0069–0.0577	0.125
τ_P-181_/Aβ_42_ vs. τ_P-181_/(Aβ_42_/Aβ_40_)	0.009	−0.0165–0.0340	0.498
AD pathology vs. non-AD pathology			
Aβ_42_ vs. Aβ_42_/Aβ_40_	0.109	0.0604–0.157	<0.001
τ_Τ_/Aβ_42_ vs. τ_T_/(Aβ_42_/Aβ_40_)	0.010	−0.0056–0.0256	0.200
τ_P-181_/Aβ_42_ vs. τ_P-181_/(Aβ_42_/Aβ_40_)	0.014	−0.0037–0.0309	0.123

**Table 4 diagnostics-13-00783-t004:** ROC curve analysis of the diagnostic accuracy of CSF biomarkers, biomarker ratios, and composite markers in the differentiation of *Alzheimer’s disease pathology* from non-Alzheimer’s disease pathologies, irrespective of the clinical phenotype. AUC: area under the curve; 95% CI: 95% confidence interval; YI: Youden Index.

	AUC	95% CI	*p*-Value	YI	Cut-Off	Sens	Spec
CSF biomarkers							
Aβ_42_	0.831	0.777–0.876	<0.0001	0.619	<430.7	95.0	66.9
τ_Τ_	0.948	0.912–0.972	<0.0001	0.845	>397	100	84.5
τ_P-181_	0.960	0.927–0.981	<0.0001	0.823	>74.8	95.0	87.3
CSF biomarker ratios							
Aβ_42_/Aβ_40_	0.939	0.901–0.966	<0.0001	0.806	<0.078	95.0	85.7
τ_P-181_/τ_T_	0.780	0.722–0.831	<0.0001	0.574	>0.164	93.3	64.1
τ_T_/Aβ_42_	0.975	0.947–0.991	<0.0001	0.901	>1.03	100	90.1
τ_P-181_/Aβ_42_	0.952	0.916–0.975	<0.0001	0.823	>0.155	98.3	84.0
CSF composite markers							
τ_T_/(Aβ_42_/Aβ_40_)/1000	0.985	0.961–0.996	<0.0001	0.912	>5.82	100	91.2
τ_P-181_/(Aβ_42_/Aβ_40_)/1000	0.965	0.934–0.985	<0.0001	0.895	>1.13	98.3	91.2

## Data Availability

The data presented in this study are available on request from the corresponding author.

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
