# Peer review of "CSF Aβ42 and Aβ42/Aβ40 Ratio in Alzheimer’s Disease and Frontotemporal Dementias"

_diagnostics, 2023, doi:10.3390/diagnostics13040783_

Round 1

Reviewer 1 Report

Dear Authors

The authors aimed to compare the diagnostic accuracy 24 of Aβ42 to Αβ42/Αβ40 ratio in: a) differentiating ADD vs. frontotemporal dementias; b) patients with 25 AD pathology vs. non-AD pathologies; c) compare biomarker ratios and composite markers to single CSF biomarkers in the differentiation of AD from FTD.

This is a quite well-executed clinical study including a relatively large number of ALS patients. The hypothesis and the study is of potential interest.
There are few limitations that must be addressed and I believe that if the authors can introduce these changes to the manuscript it would result in a significantly improved paper.

A major strength of the study that it is a quite well-executed clinical study including a relatively large number of Alzheimer patients. The hypothesis is of interest. The paper is well written in general and its message is clear.

The control group should be further described, such as whether they had any co-morbidities, average age. Patient’s Exclusion criteria should have been described.

When the authors refer"Twenty-two of the 263 subjects (8.4%; AD dementia: 16 patients; CBD: 3 patients; PSP: 1 patients; controls: 2 subjects) had discrepant results in amyloid pathology identification based on Aβ42 and Aβ42/ Αβ40 ratio and were excluded from this analysis"

What is the scientific basis for excluding analytical data?

Very small number of controls. Sample size calculations are missing from the report. Particularly for controls, there is a minimum number needed to calculate 90 % confidence limits of a 95 % reference interval determined by non-parametric statistics

In ELISA assays:

Were these samples all tested at one time (both patients and controls)? If not, what was the between-run imprecision of this test at the author's lab (not what is in the IFU, but what was verified by authors)? Where same lot numbers of kits used for the duration of the study? If not, what was done by the testing laboratory to ensure that longitudinal findings are equivalent? Merely indicating one ran controls does NOT suffice, as the controls are part of the kit which assures test run, but may not assure transparency of results generated between kit lots...unless the same lot of control material or samples from one kit lot were tested with the other.

B. What is the within-run imprecision of this test (not what is in the IFU, but what was verified by authors)?

Author Response

Reviewer 1

Comment 1: 

“The control group should be further described, such as whether they had any co-morbidities, average age. Patient’s Exclusion criteria should have been described”.

 Response:

The control group is described in Table 1. Average age was 74 (69-76) years. Data regarding education, as well as neuropsychological profiles are also included in Table 1. Regarding the characteristics  of the control subjects, these are described in the “2.1. Patients” sub-section of the “2. Materials and Methods” section:

“For comparison reasons, a control group was included. This consisted of otherwise healthy subjects undergoing knee or hip joint surgery or hernia repair under spinal anesthesia. These subjects had a negative history of cognitive or behavioral/psychiatric disorder and no clinical evidence of any major disease. All subjects had normal scores on neuropsychological testing (Mini Mental State Examination and Frontal Assessment Battery)”.

This was altered to:

“For comparison reasons, a control group was included. This consisted of otherwise healthy subjects, with no comorbidities, undergoing knee or hip joint surgery or hernia repair under spinal anesthesia. These subjects had a negative history of cognitive or behavioral/psychiatric disorder and no clinical evidence of any major disease. All subjects had normal scores on neuropsychological testing (Mini Mental State Examination and Frontal Assessment Battery)”.

Comment 2

“When the authors refer "Twenty-two of the 263 subjects (8.4%; AD dementia: 16 patients; CBD: 3 patients; PSP: 1 patients; controls: 2 subjects) had discrepant results in amyloid pathology identification based on Aβ42 and Aβ42/ Αβ40 ratio and were excluded from this analysis". What is the scientific basis for excluding analytical data?”

Response:

This study aims to compare the diagnostic accuracy of Aβ42 to Αβ42/Αβ40 ratio in the differential diagnosis of Alzheimer’s disease dementia from the behavioral variant of frontotemporal dementia. In a second analysis, we aimed to compare their diagnostic accuracy in the differentiation of Alzheimer’s disease pathology vs. non-AD pathology, irrespective of clinical phenotype.

To this end, we applied the BIOMARKAPD/ABSI criteria, which require abnormal biomarker of amyloid pathology (A), tau pathology (T) and neurodegeneration (N). Both Αβ42 and Αβ42/Αβ40 are markers of amyloid pathology. Since the aim of this study was to compare these two amyloid markers, we selected to include in the analyses only subjects with an agreement in the classification of amyloid pathology between Αβ42 and Αβ42/Αβ40. Patients with discrepant classification regarding amyloid pathology cannot be confidently classified based on the BIOMARKAPD/ABSI criteria and were thus excluded.

Comment 3:

“Very small number of controls. Sample size calculations are missing from the report. Particularly for controls, there is a minimum number needed to calculate 90 % confidence limits of a 95 % reference interval determined by non-parametric statistics”.

Response:

For the purposes of the present study, we included a sub-cohort of control subjects, with available data on all four classical CSF AD biomarkers (Αβ42, Αβ40, τT, τP-181). These subjects were included for reasons of completion, and were not used in the present study to define cut-off points.

Comment 4:

“Were these samples all tested at one time (both patients and controls)? If not, what was the between-run imprecision of this test at the author's lab (not what is in the IFU, but what was verified by authors)? Where same lot numbers of kits used for the duration of the study? If not, what was done by the testing laboratory to ensure that longitudinal findings are equivalent? Merely indicating one ran controls does NOT suffice, as the controls are part of the kit which assures test run, but may not assure transparency of results generated between kit lots...unless the same lot of control material or samples from one kit lot were tested with the other”.

Response:

Thank you for your comment on this important issue. Our laboratory implements both internal and external quality control measures to ensure the accuracy of measurements longitudinally. Specifically, for internal control a pooled CSF sample is used in every test run, resulting an over >90% between-run precision. As for external control we participate in “The Alzheimer´s Association´s QC program" which provides additional external pooled CSF samples for validating results reliability regardless of  kit's lot number.

Reviewer 2 Report

This manuscript examines the ratio of A42 and A42/40 in CSF in Alzheimer's disease and frontotemporal dementia. In order to find the significance of this manuscript, it is necessary to clarify the difference between this manuscript and the previous ones. If there is a difference in the kit used for the immediate composition, there is a need to clarify the superiority of the kit.

The authors should improve the discussion section more and resubmit the manuscript.

Author Response

Reviewer 2

Comment 1:

“This manuscript examines the ratio of A42 and A42/40 in CSF in Alzheimer's disease and frontotemporal dementia. In order to find the significance of this manuscript, it is necessary to clarify the difference between this manuscript and the previous ones. If there is a difference in the kit used for the immediate composition, there is a need to clarify the superiority of the kit”.

Response:

The ratio of Αβ42/Αβ40 compared to Αβ42 has indeed been studied previously. However, most of these studies related to the differential diagnosis of Alzheimer’s disease from healthy subjects, or looked into the predictive value of these markers in the evolution of healthy subjects to mild cognitive impairment or from MCI to AD.

Few studies have examined these markers in cohorts of different neurodegenerative dementias. In these studies, the diagnosis was based on clinical criteria solely. The novelty of our study lies in the comparison of Aβ42 vs. Aβ42/Αβ40 on two levels: a) a clinical level (in accordance with all previous relative studies), wherein a clinical diagnosis of Alzheimer’s disease dementia (ADD), behavioural variant FTD (bvFTD), progressive supranuclear palsy (PSP) and corticobasal syndrome (CBS) is set, and these phenotypes are compared, and: b) a pathological level, wherein the BIOMARKAPD/ABSI are applied to classify all patients into Alzheimer’s disease and non-Alzheimer’s disease pathology based on their CSF biomarkers. This analysis in essence is a comparison between AD pathology and frontotemporal lobar degeneration (FTLD).

This is described in detail in the 2.5 “Statistical analysis”subsection:

“The second analysis aimed to investigate the diagnostic accuracy of single CSF bi-omarkers, biomarker ratios and composite biomarkers in identifying AD pathology ir-respective of the clinical phenotype. To this end, a two-step process was applied, as described elsewhere.

Initially, CSF biomarkers were transformed into binary variables (i.e. normal or ab-normal), based on cut-off values of the Unit of Neurochemistry and Biomarkers (Aβ42 <480pg/ml; τT>400pg/ml; τP-181>60pg/ml; Aβ42/Aβ40<0.094), as described previously[34]. Subjects with abnormal Αβ42, τT, τP-181 and Aβ42/Αβ40 ratio (A+T+N+) were considered to harbor AD pathology, based on the BIOMARKAPD/ABSI criteria[26, 27]. Thus, all patients, irrespective of their clinical phenotype were classified into two categories: AD vs. nonAD”.

The following paragraph has been added in the Statisticla analysis subsection, to clarify this point:

Thus, this second analysis included 82 patients with AD pathology and 140 patients with non-AD pathology: 49 bvFTD patients, 42 CBD patients and 49 PSP patients. The majority of the non-AD pathology patients are considered to have an underlying frontotemporal lobar degeneration (FTLD), with most PSP and CBD patients harboring an FTLD with tau proteinopathy (FTLD-tau) and most bvFTD patients harboring either an FTLD-tau or FTLD-TDP43 proteinopathy. Thus, in essence the second analysis referred to comparison of Aβ42 to Aβ42/Αβ40 ratio in the differentiation of AD pathology from FTLD. However, due to the lack of neuropathological confirmation, we elected to use the term non-AD pathology instead of FTLD, since the presumed underlying pathology is based on CSF biomarker profiles”.

Comment 2:

“The authors should improve the discussion section more and resubmit the manuscript”.

Response:

We have altered the discussion section to clarify the importance of CSF biomarkers in differentiating AD pathology from FTLD.

Reviewer 3 Report

This manuscript by Constantinides et al. aims to compare the diagnostic accuracy of AB42 levels in the cerebral spinal fluid (CSF) to the AB42/AB40 ratio in the CSF. They want to see if either can differentiate between Alzheimer’s disease dementia (ADD) and other frontotemporal dementias (FTDs). They also want to see if either can differentiate between patients with Alzhemier’s disease (AD) and non-AD pathologies. They concluded that AB42 levels did not differ from the AB42/AB40 ratio in differentiating between ADD and FTDs, but the AB42/AB40 ratio was superior to AB42 in differentiating between AD pathology and non-AD pathology. There were two significant problems with this paper. First, it was unclear how they defined non-AD pathology.  Since the first comparison was between ADD and other FTDs, it is assumed that non-AD pathology meant healthy controls, but this was confusing.  Second the observation that the AB42/AB40 ratio was superior to AB42 in differentiating between AD pathology and non-AD pathology is not a novel finding.  There are already many papers (including a review summarizing them all that was cited in this paper) that make the same finding and come to the same conclusion.  Perhaps, a novel finding from this paper would have been if they could find a way to use AB42 or the AB42/AB40 ratio to differentiate ADD from other FTDs, but they were not able to do that.  I do not recommend publication of this paper.

Author Response

Reviewer 3

Comment 1:

 “First, it was unclear how they defined non-AD pathology.  Since the first comparison was between ADD and other FTDs, it is assumed that non-AD pathology meant healthy controls, but this was confusing”.

Response:

Thank you for your helpful comment. In the second analysis, AD pathology vs. non-AD pathology was defined after binarization of CSF biomarkers and application of the BIOMARKAPD/ABSI criteria. These criteria define AD pathology as the presence of abnormal values in all three major biomarker groups: Aβ42 or Αβ42/Αβ40 ratio for amyloid pathology (A); τP-181 for tau pathology (T); τT for neurodegeneration (N). Thus, according to BIOMARKAPD/ABSI criteria, only A(+)T(+)N(+) patients are considered as patients with underlying AD pathology. All other patients are considered as harbouring a non-AD pathology.

This is described in the statistical analysis section:

“The second analysis aimed to investigate the diagnostic accuracy of single CSF biomarkers, biomarker ratios and composite biomarkers in identifying AD pathology irrespective of the clinical phenotype. To this end, a two-step process was applied, as described elsewhere.

Initially, CSF biomarkers were transformed into binary variables (i.e. normal or abnormal), based on cut-off values of the Unit of Neurochemistry and Biomarkers (Aβ42 <480pg/ml; τT>400pg/ml; τP-181>60pg/ml; Aβ42/Aβ40<0.094), as described previously[34]. Subjects with abnormal Αβ42, τT, τP-181 and Aβ42/Αβ40 ratio (A+T+N+) were considered to harbor AD pathology, based on the BIOMARKAPD/ABSI criteria[26, 27]. Thus, all patients, irrespective of their clinical phenotype were classified into two categories: AD vs. nonAD”.

We elected to apply the most stringent classification criterion (i.e. all three CSF biomarkers abnormal) for AD pathology identification in order to increase specificity, at the expense of sensitivity. For amyloid pathology (A) in particular, there were two available established CSF markers: Αβ42 and the Αβ42/Αβ40 ratio. In an effort to increase specificity, subjects with a decrease in both markers (Αβ42 and Αβ42/Αβ40) were considered A(+), and only subjects with normal values in both biomarkers were considered A(-).

Comment 2:

“Second, the observation that the AB42/AB40 ratio was superior to AB42 in differentiating between AD pathology and non-AD pathology is not a novel finding.  There are already many papers (including a review summarizing them all that was cited in this paper) that make the same finding and come to the same conclusion.  Perhaps, a novel finding from this paper would have been if they could find a way to use AB42 or the AB42/AB40 ratio to differentiate ADD from other FTDs, but they were not able to do that.  I do not recommend publication of this paper”.

Response:

Thank you for your insightful remark. Indeed, there are abundant papers that have supported the superiority of the Aβ42/Αβ40 ratio over Aβ42 in the differentiation of AD from healthy subjects. In our paper, we have concluded that the Aβ42/Αβ40 ratio is superior to Ab42 in the differentiation of patients with an AD pathology vs. non-AD pathology.

The cohorts of this analysis (after exclusion of 22 of the 263 subjects: AD dementia: 16 patients; CBD: 3 patients; PSP: 1 patients; controls: 2 subjects) included: 82 AD patients, 49 bvFTD patients, 42 CBD patients, 49 PSP patients.

To sum up, the patients with presumed AD pathology were 82, and patients with presumed non-AD pathology were 140 in total (49+42+49). All of the patients with a non-AD pathology probably have and an underlying frontotemporal lobar degeneration (FTLD) either with tau proteinopathy (FTLD-tau: PSP, CBD and ~50% of bvFTD) or a TDP-43 proteinopathy (50% of bvFTD).

Thus, the present study does indeed support that the Aβ42/Αβ40 ratio is superior to Ab42 in differentiating AD pathology from FTLD. The reason we have selected to use the term non-AD pathology instead of FTLD throughout the manuscript, is that our cohort lacks neuropathological confirmation, and the classification of presumed underlying pathology is based on CSF biomarker profiles.

The following paragraph has been added in the Statistical analysis subsection to clarify this point:

Thus, this second analysis included 82 patients with AD pathology and 140 patients with non-AD pathology: 49 bvFTD patients, 42 CBD patients and 49 PSP patients. The majority of the non-AD pathology patients are considered to have an underlying frontotemporal lobar degeneration (FTLD), with most PSP and CBD patients harboring an FTLD with tau proteinopathy (FTLD-tau) and most bvFTD patients harboring either an FTLD-tau or FTLD-TDP43 proteinopathy. Thus, in essence the second analysis referred to comparison of Aβ42 to Aβ42/Αβ40 ratio in the differentiation of AD pathology from FTLD. However, due to the lack of neuropathological confirmation, we elected to use the term non-AD pathology instead of FTLD, since the presumed underlying pathology is based on CSF biomarker profiles”.

Round 2

Reviewer 2 Report

The revised manuscript, on the contrary, turned out to be very bad, and I regret that as a reviewer. Since there have been many manuscripts on the comparison of AD and non-AD in spinal fluid, we focused our assay and analysis on tauopathy and MND, for example, among neurodegenerative diseases. On the other hand, how did the results of neurodegenerative diseases compared to AD? I wrote a comment on the peer review to emphasize that I wanted to emphasize that the results were not the same, but it turned out to be quite the opposite, which is unfortunate. I hope you will submit the manuscript again in the form of an assay and analysis focusing on tauopathy and MND, for example, in neurodegenerative diseases.

Author Response

Reply to Reviewer 2

Comment 1:

“The revised manuscript, on the contrary, turned out to be very bad, and I regret that as a reviewer. Since there have been many manuscripts on the comparison of AD and non-AD in spinal fluid, we focused our assay and analysis on tauopathy and MND, for example, among neurodegenerative diseases”.

Reply:

We respect the concerns raised by the reviewer, however, our paper does not include patients with MND or ALS, the most common of the MND syndromes. ALS is indeed a TDP43 proteinopathy in 97% of the cases, which manifest with upper and lower motor neuron dysfunction, thus it does not enter in the differential diagnosis of dementias, if they are not accompanied by FTD. Our patient groups as described in the “Patients” subsection of the Methods section include patients with behavioral variant frontotemporal dementia, progressive supranuclear palsy, corticobasal syndrome (under the umbrella of FTD) and Alzheimer’s disease dementia.

The only modifications we made to the initial manuscript, compared to the revised, were the addition of two paragraphs explaining the presumed underlying pathologies of bvFTD, PSP and CBD (patients with bvFTD have either FTLD with tau or FTLD with TDP43 proteinopathy, whereas CBS and PSP patients, most frequently, have an underlying tauopathy), as stated in

Comment 2:

“On the other hand, how did the results of neurodegenerative diseases compared to AD?”

Reply:

The results regard the comparison of Aβ42 vs. Αβ42/Αβ40 ratio in AD vs. the presumed FTLD cohort on clinical grounds (which however has significant overlap on pathological ground as stated in the Results section) in an effort to investigate a possible superiority of the ratio in the discrimination of the two groups. Tables 1-4 and Figures 1-2 of the manuscript further present these results.

Comment 3:

“I wrote a comment on the peer review to emphasize that I wanted to emphasize that the results were not the same, but it turned out to be quite the opposite, which is unfortunate. I hope you will submit the manuscript again in the form of an assay and analysis focusing on tauopathy and MND, for example, in neurodegenerative diseases”.

Reply:

We would like to re-emphasize that the present manuscript has no relevance to MND (or ALS), which is the reason why MND is not mentioned throughout the manuscript.

Reviewer 3 Report

Authors have addressed my comments, though I still feel the novelty of these experiments is very low.

Author Response

Response to Reviewer 3

Comment:

“Authors have addressed my comments, though I still feel the novelty of these experiments is very low”.

Reply:

Thank you for your review.
